# Human-environment feedback and the consistency of proenvironmental behavior

Claire Ecotière[1]*, Sylvain Billiard[2], Jean-Baptiste André[3], Pierre Collet[4], Régis Ferrière[5,6,7‡], Sylvie Méléard[1,8‡]

1 Centre de Mathématiques Appliquées, CNRS, Ecole Polytechnique, IP Paris, Palaiseau, France, 2 Evo-Eco-Paleo, Univ. Lille, CNRS, Lille, France, 3 Institut Jean Nicod, Département d'études cognitives, ENS, EHESS, PSL Research University, CNRS, Paris France, 4 CPHT, CNRS, Ecole polytechnique, IP Paris, Palaiseau, France, 5 Institut de Biologie (IBENS), ENS-PSL, CNRS, INSERM, Paris, France, 6 Ecology and Evolutionary Biology, University of Arizona, Tucson, Arizona, United States of America, 7 iGLOBES International Research Laboratory, CNRS, ENS-PSL, University of Arizona, Tucson, Arizona, United States of America, 8 Institut Universitaire de France, Paris, France

‡These authors are joint senior authors on this work.
* claire.ecotiere@polytechnique.edu

**Data Availability Statement:** All relevant data are within the manuscript and its Supporting information files.

## Abstract

Addressing global environmental crises such as anthropogenic climate change requires the consistent adoption of proenvironmental behavior by a large part of a population. Here, we develop a mathematical model of a simple behavior-environment feedback loop to ask how the individual assessment of the environmental state combines with social interactions to influence the consistent adoption of proenvironmental behavior, and how this feeds back to the perceived environmental state. In this stochastic individual-based model, individuals can switch between two behaviors, 'active' (or actively proenvironmental) and 'baseline', differing in their perceived cost (higher for the active behavior) and environmental impact (lower for the active behavior). We show that the deterministic dynamics and the stochastic fluctuations of the system can be approximated by ordinary differential equations and a Ornstein-Uhlenbeck type process. By definition, the proenvironmental behavior is adopted consistently when, at population stationary state, its frequency is high and random fluctuations in frequency are small. We find that the combination of social and environmental feedbacks can promote the spread of costly proenvironmental behavior when neither, operating in isolation, would. To be adopted consistently, strong social pressure for proenvironmental action is necessary but not sufficient—social interactions must occur on a faster timescale compared to individual assessment, and the difference in environmental impact must be small. This simple model suggests a scenario to achieve large reductions in environmental impact, which involves incrementally more active and potentially more costly behavior being consistently adopted under increasing social pressure for proenvironmentalism.

## Author summary

Reducing global environmental degradation such as climate warming requires the adoption of consistent proenvironmental behaviors. But we, as individuals, tend to act in

**Funding:** This work was funded by a grant from the program 80PRIME of the French National Center for Scientific Research (CNRS). Funding was also provided by the iGLOBES Mobility Program of Paris Sciences & Lettres University and was partially funded by the Chair "Modélisation Mathématique et Biodiversité" of VEOLIA-Ecole Polytechnique-MNHN-F.X. JBA acknowl- edges support from the EUR FrontCog grant ANR-17-EURE-0017. The funders had no role in study design, data collection and analysis, decision to publish, or preparation of the manuscript.

**Competing interests:** None of the authors have a competing interests that could be perceived to bias this work.

response to alarming events, and relax when things seem to get better. Can this tendency to behave inconsistently be countered by social interactions and social pressure? We explore this question by developing a simple mathematical model. We find that social interactions, in combination with our own individual perception of population's environmental impact, can indeed promote the consistent adoption of costly environmental behavior, but only when the social pressure for proenvironmentalism is strong enough. Our model suggests a possible path through 'small steps' from 'business as usual' behavior to active behavior with large reduction of the environmental impact. In the small-steps scenario, a gradual buildup of social pressure for proenvironmentalism can lead to a large reduction in perceived environmental impact even if the active behaviors adopted by the population become incrementally more costly. Thus, creating contexts that are conducive to strong social pressure for proenvironmentalism, through communication and public policy, appears both critical and powerful to achieve behavioral change that can make a difference.

## Introduction

Why don't we all act more decisively in the face of global environmental crises such as climate change or biodiversity loss? Achieving climate and biodiversity targets set by international agreements (e.g. Paris accord, Aichi convention) ultimately requires consistent behavioral changes within and across societies. At the level of individuals, limiting climate change or biodiversity loss requires consistent consumer choices with reduced net environmental impact. As citizens, individuals must consistently promote governmental policies that favor proenvironmental actions. Leaders and senior managers, as individuals, should make consistent decisions to influence greenhouse gas emissions and natural resource use by large organizations and industries.

For many individuals, adopting a proenvironmental behavior is not straightforward. Indeed the decision amounts to accepting certain short-term costs and reductions in living standards in order to mitigate against higher but uncertain losses that may be far in the future [1]. Individual behavioral responses to this collective-risk social dilemma [2] are not all-or-nothing, however. Between those who unconditionally accept or unconditionally deny the need for action towards environmental sustainability, the vast majority of people do not engage consistently in either way. Rather, non-ideologically polarized individuals tend to show inconsistent behavior as they change opinion, revise their intention, or switch behavior during their lifetime [3], possibly on very short timescales [4].

Such a behavioral inconsistency can be due to various mechanisms. Individuals who engage in some kind of proenvironmental action may lose motivation to "take the next step". In this case, action limits intention for more, a pattern called *tokenism* [1]. In the same vein, the *rebound effect* occurs when some mitigating effort is diminished or erased by the individual's subsequent actions [5]. For example, after acquiring a more fuel-efficient vehicle (an active mitigating behavior), owners tend to drive them farther, in effect reverting to their baseline environmental impact [6]. Other patterns of inconsistent behavior involve responses to extreme climatic events. Exposure to a climate-related hazard such as wildfires increases support for costly, pro-climate ballot measures in subsequent local elections [7]; but the degree of personal concern about climate change is related to the temperature anomaly only over a few months in the past [8]. Thus, outside of the most politically polarized groups, the influence of environmental anomalies can be strong, but it decays rapidly [9].

Mathematical modeling of human-environment dynamics can help identify pathways toward proenvironmental behavior consistency. Game-theoretic models with environmental feedbacks, or eco-evolutionary games, have been developed to study coupled environmental and behavior dynamics [10–14]. Specific examples address the use of a natural resource such as farmland [15], water [16] or forest [17]. A key aspect of these behavior-environment models [12, 18, 19] based on 'imitation dynamics' [20] is that individuals' behavioral decisions are only made in the context of their interaction with others. Yet factors that are not tied to social encounters, such as the individual's own experience and perception of the environmental state, may play a key role in environmental decisions. This is core to the conceptual framework that Schill et al. (2019) [21] recently put forth, building on behavioral economics and cognitive psychology, for understanding human behavior in the face of sustainability challenges. In this framework, individuals create social and environmental contexts that change dynamically with continuous feedback to their behavior. Following Schill et al. (2019) [21] we hypothesize that the dynamics of environmental behavior are shaped both by social context and the individuals' private environmental experience.

Here we implement this hypothesis by constructing a simple mathematical stochastic model based on individual-level rules. Each individual has a negative impact on their environment that depends on their behavior—the impact of an active, proenvironmental behavior being less than the impact of the baseline behavior. Individuals can change their behavior in response to both social interactions and their own perception of environmental degradation. The environmental state and the individuals' behavior are modeled as continuous and discrete variables, respectively, and the different processes affecting the state of the behavior-environment system play out on different time scales. We use the model to investigate the determinants of proenvironmental behavior consistency. We say that a behavior is adopted consistently when its frequency at stationary state is high and the stochastic fluctuations in frequency are small. Our model analysis addresses how the individual assessment of environmental degradation combines with social interactions to determine the consistent adoption of proenvironmental behavior, and how this feeds back to the perceived environmental state. In particular, we ask whether larger costs of, or weak social pressure on, proenvironmental behavior make the consistent adoption of proenvironmental behavior less likely; and whether a slower pace of change in the perceived environmental state can promote consistency.

## Results

### Model overview

We consider a population of $N$ agents that interact among themselves and perceive the state of the environment (more or less degraded) through a single environmental variable, $e$. Individual behavior and their environmental impact are modeled on a short enough timescale so that $N$ remains constant. The $e$ variable measures the perceived environmental state on a continuous scale, with larger $e$ corresponding to an environment perceived as more degraded. The $e$ variable can be seen as an indicator or summary statistics of the perceived level of environmental degradation, whose variation is driven by the population level of environmental action, intention, or awareness, such as the spread of renewable energy, the adoption of plant-based diets, the reduced consumption of non-essential goods, or the prominence of pro-environmental demonstrations and other public calls for proenvironmental action. At any time $t$ each individual can express one of two behaviors: baseline (denoted by $B$) or active (denoted by $A$). When expressing behavior $A$, an agent actively seeks to reduce their negative environmental impact compared to the baseline impact of behavior $B$. This is modeled by assuming that an agent in state $A$ increases the perceived environmental impact of the population by an amount

$l_A$, which is less than the environmental impact, $l_B$, of behavior $B$ (per capita). We also assume that $l_A$ is positive meaning that the population influences the environment by its own existence. Agents can switch between behaviors $A$ and $B$.

In the Methods we expound the mathematical derivation of a model for the joint dynamics of the frequency of behavior $A$ and the perceived environmental degradation. Notations are listed in Table 1. We assume that the dynamics of the perceived environnemental state follows a deterministic continuous process. We define parameter $\ell$ as representing the timescale at which individuals' behavior affects the perceived environmental state: the higher $\ell$, the faster the perceived environmental state changes due to individuals' behavior. In a population where all agents express behavior $A$ ($B$, respectively), the rate of change of the environment perceived as minimally (maximally) degraded is proportional to $l_A$ ($l_B$) and the stationary value of the perceived environmental state is $l_A$ ($l_B$). In a population where both behaviors are expressed, the perceived environmental state varies between $l_A$ and $l_B$.

Any agent may switch at any time between behaviors $A$ and $B$ as a result of encounters with other agents (modeling social interactions). We define $\kappa$ as a scaling parameter controlling the rate of switching behavior via social interactions (modeled as encounters with other agents). The rate at which an agent changes its behavior upon encountering another agent depends on the attractiveness of the alternate behavior, which is determined by the perceived payoff differential between the two behaviors, and the social pressure. We use parameters $\gamma_i$, $i = A$ or $B$, the payoff from adopting behavior $i$, and $\delta_i$, the social pressure for behavior $i$. We denote the payoff difference between behaviors $A$ and $B$, or payoff differential, by $\beta = \gamma_A - \gamma_B$. We say that the active behavior $A$ is *costly* when the payoff differential, $\beta$, is negative. The payoff differential may be positive if, for example, the active behavior $A$ is actually incentivized through public policy.

Any agent may also switch behavior at any time based on their assessment of the environmental state. Agents tend to adopt the alternate behavior when they perceive the

**Table 1. Parameter definitions and default values.**

| Notation | Parameter description | Default value |
|---|---|---|
| $A$, $B$ | Active vs. Baseline behaviors | |
| $N$ | Size of the population | |
| $N_t^{A,N}$, $N_t^{B,N}$ | Number of individuals expressing behavior $A$ or $B$ at time $t$ | |
| $X_t^N$; $E_t^N$ | Frequency of individuals with behavior $A$; perceived environmental state at time $t$ (dimensionless) | |
| $x_t$; $e_t$ | Deterministic frequency of individuals with behavior $A$; deterministic perceived environmental state at time $t$ (dimensionless) | |
| $\kappa$ | Encounter rate (inverse of unit time) | 1 |
| $\tau$ | Individual sensitivity to the environment (inverse of unit time) | |
| $\ell$ | Environmental reactivity (inverse of unit time) | |
| $\gamma_A$ (resp. $\gamma_B$) | Payoff of behavior $A$ (resp. $B$) (dimensionless) | $\gamma_B = 1$ |
| $\delta_A$ (resp. $\delta_B$) | Social pressure of behavior $A$ (resp. $B$) (dimensionless) | $\delta_B = 0.5$ |
| $l_A$ (resp. $l_B$) | Individual environmental impact of behavior $A$ (resp. $B$) (dimensionless) | $l_B = 1$ |
| $\beta = \gamma_A - \gamma_B$ | Payoff differential (dimensionless) | |
| $\frac{\delta_B}{\delta_A + \delta_B}$ | Social norm threshold (SNT) (dimensionless) | |
| $l_B - l_A$ | Environmental impact differential (dimensionless) | |

environmental impact of their current behavior as relatively large compared to the alternate behavior. We use parameter $\tau$ to set the timescale of behavioral switch based on individual assessment.

The dynamics of the coupled behavior-environment process are stochastic, driven by the probabilistic events of behavior switch between $A$ and $B$, under the joint effects of encounters among agents and individual assessment, and the deterministic response of the perceived environmental state. The mathematical model we obtain is a Piecewise Deterministic Markov Process where the population state (frequencies of behaviors) probabilistically jumps at each change in individual behavior while the environmental state deterministically and continuously changes between jumps. Note that the model assumes that the perceived environmental state and social context are the same for all individuals, and individuals do not differ in personality, in the sense that all individuals have the same intrinsic propensity to change their behavior (or not) across time.

## Deterministic approximation of the dynamics and stochastic fluctuations

When the population size $N$ is very large and under assumptions on the rates stated below, the behavior-environment dynamics can be approximated (in a sense made rigorous in the Methods and S1 Appendix) by the unique solution of the following system of ordinary differential equations for the active behavior frequency, $x$, and perceived environmental state, $e$,

$$\frac{dx_t}{dt} = p(x_t, e_t) = \kappa x_t (1 - x_t)(\lambda_A(x_t) - \lambda_B(x_t)) + \tau_A(e_t)(1 - x_t) - \tau_B(e_t)x_t$$

$$\frac{de_t}{dt} = h(x_t, e_t) = \ell e_t(l_A x_t + l_B(1 - x_t) - e_t)$$

(1)

with initial conditions $(x_0, e_0)$ in $[0, 1] \times [l_A, l_B]$. The first equation governs the frequency $x$ of the active behavior, $A$. In the right hand side of $p(x, e)$, the first term measures behavior switching due to encounters (social interactions), with $B$ switching to $A$ at rate $\lambda_A(x)$ and $A$ switching to $B$ at rate $\lambda_B(x)$, given by

$$\lambda_A(x) = \gamma_A + \delta_A x$$

$$\lambda_B(x) = \gamma_B + \delta_B(1 - x)$$

(2)

(following from Eqs (10), (11) and (8), see Methods). The second term measures behavior switching due to individual assessement of the environmental state. The corresponding rates of switching from $B$ to $A$ and from $A$ to $B$ respectively are

$$\tau_A(e) = \tau(e - l_A)$$

$$\tau_B(e) = \tau(l_B - e)$$

(3)

(following from Eq (13) in the Methods). The second equation in System (1) drives the dynamics of the perceived environmental state, $e$. An equilibrium point $(x^*, e^*)$ is a $(x, e)$ pair that nullifies both functions $p(x, e)$ and $h(x, e)$.

Even though the rates of behavior switching are deterministic functions, the individual switching events occur probabilistically. As a consequence, given that the population size is finite, the actual frequency of the behaviors fluctuates randomly, even asymptotically around the equilibrium values predicted by the deterministic model. In the Methods and S1 Appendix, we show that the variance of the random fluctuations in frequency is equal to the total rate of behavior switching between $A$ and $B$, elicited by social interactions and individual assessment

of the environmental state. This total rate, denoted by $\Omega$, is equal to

$$\Omega(x, e) = \kappa x(1 - x)(\lambda_A(x) + \lambda_B(x)) + \tau_A(e)(1 - x) + \tau_B(e)x \tag{4}$$

given the values of the $A$ frequency, $x$, and perceived environmental state, $e$. The total rate $\Omega$ measures the intensity of switching between behaviors across the population. Thus, the higher $\Omega$, the larger the probability that some agent switches behavior. We characterize $A$ as being consistently adopted when the $A$ equilibrium frequency, $x^*$, is high (close to one) and the corresponding total rate of behavior switching, $\Omega(x^*, e^*)$, is low (close to zero).

In the following subsections we first describe the deterministic dynamics of the large-population model in the absence of environmental feedback ($\tau = 0$ in Eq (1)). When the environmental feedback is included, we investigate the effect of all parameters to identify those that control the value and stability of equilibria: payoff differential, $\beta$, social norm threshold, $v = \frac{\delta_B}{\delta_A + \delta_B}$, individual environmental impacts and environmental impact differential, $l_B - l_A$, individual sensitivity to the environmental state, $\tau$, and reactivity of the environment, $\ell$. We then identify conditions for the spread and consistent adoption of the $A$ behavior. Finally, we qualitatively discuss how incremental variation in the proenvironmental behavior could affect the behavior-environment system dynamics and lead to a robust reduction of the individuals' perceived environmental impact.

## Behavior dynamics without environmental feedback

In the absence of environmental feedback (i.e. no individual assessment, $\tau = 0$), agents may switch behavior only upon encountering other agents, i.e. through social interactions. Eq (1) then reduces to the standard imitation dynamics (or replicator) equation

$$\begin{aligned}
\frac{dx_t}{dt} &= \kappa x_t(1 - x_t)[\beta + \delta_A x_t - \delta_B(1 - x_t)] = p_0(x_t), \\
\frac{de_t}{dt} &= h(x_t, e_t) = \ell e_t(l_A x_t + l_B(1 - x_t) - e_t)
\end{aligned} \tag{5}$$

Assuming $l_A > 0$ and $e_0$ between $l_A$ and $l_B$, the only possible equilibrium value for the environmental state is $e^* = l_A x^* + l_B(1 - x^*)$. The model admits three equilibria, $x_0^* = 0$, $x_1^* = 1$ and $x^* = \frac{\delta_B - \beta}{\delta_A + \delta_B}$. If $x^* < 0$ (resp. $x^* > 1$), then $x_1^* = 1$ (resp. $x_0^* = 0$) is globally stable. If $0 < x^* < 1$, then the system is bistable; convergence to $x_1^* = 1$ occurs if the initial frequency of the active behavior is higher than $x^*$ (Fig 1). Note that when the payoff differential $\beta$ is null, the outcome is entirely determined by social pressures and in this case, the frequency threshold $x^*$ is equal to the social norm threshold, $v = \frac{\delta_B}{\delta_A + \delta_B}$.

At stable equilibrium ($x_0^* = 0$ or $x_1^* = 1$) the total rate of behavior switching, $\Omega$ (Eq (4)), is always equal to zero, which means that the prevailing behavior, $A$ or $B$, is adopted consistently. Thus, if the active behavior $A$ spreads, it will necessarily be adopted consistently; but the spread of $A$ from low frequency requires $A$ to be perceived as sufficiently rewarding compared to $B$ (i.e.$\gamma_A > \gamma_B + \delta_B$).

## Effect of environmental feedback on active behavior frequency

As expected, the environmental feedback alone can prevent the active behavior from spreading to high frequency. By taking $\tau > 0$ and $\kappa = 0$ in Eq (1), individual behavior is influenced by the perceived environmental state and not by social interactions. In this case, Eq (1) possesses only one stable equilibrium, $(x^*, e^*) = \left(\frac{1}{2}, \frac{l_B + l_A}{2}\right)$. With no social interactions ($\kappa = 0$) the payoffs $\gamma_A$

and $\gamma_B$ have no influence on the equilibrium frequency of the active behavior, since the payoffs only play a role in the behavior dynamics when agents can compare them, which requires contact.

By setting both $\tau > 0$ and $\kappa > 0$ in Eq (1), the effect of environmental feedback combines with the effect of social interactions. As in the case without environmental feedback (cf. previous subsection), the model predicts up to three internal equilibria (i.e. $0 < x^* < 1$, see Methods). The stability analysis shows that the product of individual sensitivity to the environment, $\tau$, and environmental impact differential, $l_B - l_A$, is a key determinant of the system dynamics. When $\tau(l_B - l_A)$ is small enough, there is one (globally stable) or three (two stable, one unstable) equilibria, depending on the payoff differential, $\beta$, and social norm threshold, $v$ (Fig 2A–2C). The stable equilibrium is always close to $x^* = 0$ or $x^* = 1$ while the two stable equilibria are close to $x^* = 0$ and $x^* = 1$, respectively. When the product $\tau(l_B - l_A)$ is large enough, there is only one equilibrium. For a given payoff differential, $\beta$, and social norm threshold, $v$, this equilibrium can be stable or unstable (here necessarily a limit cycle) depending on environmental reactivity, $\ell$ (Fig 2D–2F). Thus, parameter $\ell$, the environmental reactivity, does not affect the number of equilibria but it affects their stability (Fig 2D–2F).

The environmental feedback thus has two main consequences for the spread of the active behavior $A$ from low frequency. On the one hand, under the condition that the product $\tau(l_B - l_A)$ is small enough (Fig 2A–2C), the increase of $A$ frequency from very low to an equilibrium close to 1 becomes possible even if $A$ is costly ($\beta < 0$) or at least low-incentivized (small $\beta > 0$), provided that the environmental feedback be combined with strong social pressure for the active behavior (low $v$ due to large $\delta_A$, Fig 2A–2C to be compared with Fig 1). On the other hand, for large values of the product $\tau(l_B - l_A)$, the environmental feedback dominates the effect of social conformism for the active behavior. As a consequence, most active behaviors can reach a frequency close to 0.5 irrespective of their cost and the intensity of social pressure (Fig 2G–2I). High frequency (close to one) can be reached but only under very strong social pressure, even for active behavior that are strongly beneficial (large $\beta > 0$).

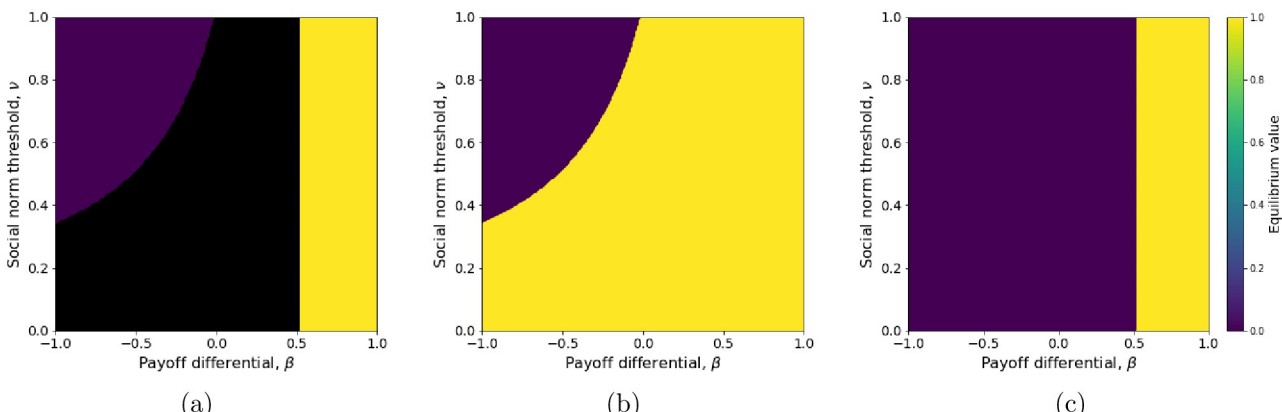

(a)                                    (b)                                    (c)

**Fig 1. Frequency of active behavior at equilibrium in the absence of environmental feedback (Eq (5)), with respect to the payoff differential ($\beta$) and social norm threshold ($v = \frac{\delta_B}{\delta_A + \delta_B}$).** (a) Bistability occurs in the black filled area (depending on the initial conditions, the equilibrium is either $x^* = 0$ or $x^* = 1$). (b) The upper equilibrium value ($x^* = 1$) is plotted across the bistability area (reachable for initial frequency $x_0 > v$). (c) The lower equilibrium value ($x^* = 0$) is plotted across the bistability area (reachable for initial frequency $x_0 < v$). Environmental sensitivity is $\ell = 0.1$ and other parameters are set to their default values (Table 1).

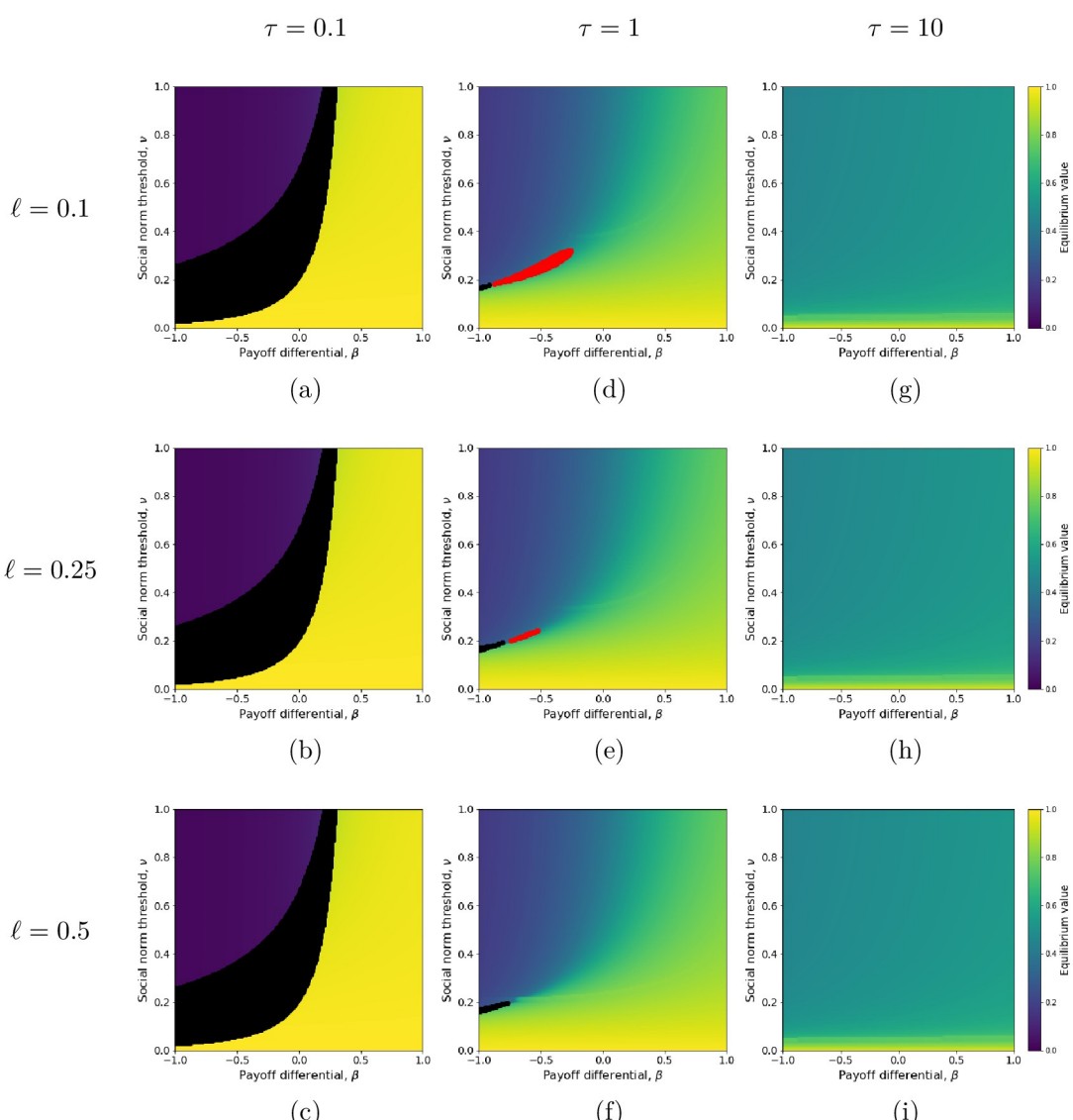

**Fig 2. Frequency of active behavior at equilibrium in the presence of environmental feedback (Eq (16)), with respect to the payoff differential ($\beta$) and social norm threshold ($v = \frac{\delta_B}{\delta_A + \delta_B}$), for low to high individual sensitivity to the environment ($\tau$) (for (a), (d) and (g) $\ell = 0.1$) and environmental reactivity ($\ell$).** The place of each panel (a)-(i) gives the values taken for $\tau$ and $\ell$ = 0.25. For example, for (e), $\ell = 0.25$ and $\tau = 1$. Bistability occurs in the black filled areas. Stable limit cycles occur in the red filled areas. The environmental impact differential is fixed ($l_B = 1$, $l_A = 0.7$). Other parameters ($\kappa$, $\delta_B$) are set to their default values (Table 1).

The spread from low to high frequency of a *costly* active behavior ($\beta < 0$) cannot occur in the absence of environmental feedback. The effect of the environmental feedback can be explained as follows. In a population where the active behavior is rarely expressed ($x$ close to zero initially), the perceived environmental state is essentially set by the baseline behavior $B$. Thus, the environment is perceived as strongly degraded, the environmental feedback promotes the behavioral switch from $B$ to $A$, hence the frequency of $A$ rises. If the social pressure of individuals expressing behavior $A$ is strong enough (i.e., if $\delta_A$ is sufficiently larger than $\delta_B$) the initial pull of the $A$ frequency by the environmental feedback will be sufficient to drive it

above the social norm threshold, $v$. Once the $A$ frequency exceeds the $v$ tipping point, social interactions pick up and drive $A$ toward its equilibrium frequency, close to 1.

Simulations of trajectories in the case of the active behavior rising from low to high frequency allow us to constrain the model unit time (Fig E in S1 Appendix). Our approach assumes a timeline over which the global environmental state or trend of global environmental degradation (e.g. climate warming) is essentially unaltered by individual behaviors. The timescale over which individuals interact, perceive the environmental state, and change their behavior is set by parameters $\kappa$, $\tau$, and $l$. With $\kappa = 1$ (i.e. one social interaction about the environmental concern expected on average per unit time), the characteristic time for the frequency of proenvironmental behavior to rise from near zero to near one is of the order of 1–50 unit time (Fig E in S1 Appendix). Thus, environment-related social encounters that happen on average once a week or once a month would be consistent with the typical dynamics of the model. With a one-week time unit, the individual assessment of the environment would occur on average every three months with $\tau = 0.1$, or roughly every day with $\tau = 10$. With a one-month time unit, individual assessment of the environment would occur, on average, roughly every year with $\tau = 0.1$, and every three days with $\tau = 10$. Over such timescales, convergence to the stationary state occurs well before the actual physical environment or environmental trend (e.g. speed of warming) might change as a consequence of the population consistently adopting proenvironmental behaviors.

## Spread and consistent adoption of a costly active behavior

The consistent adoption of a costly active behavior requires the equilibrium frequency, $x^*$, to be close to one and the total switching rate, $\Omega$ (Eq (4)), to be close to zero. Figs 2 and 3 illustrate conditions under which this is the case. According to the previous subsection, a general condition for a costly (or low-incentivized) active behavior to spread from low to high frequency ($x^*$ close to 1) is that the social pressure for $A$ be strong enough (large $\delta_A$ hence small $v$, Fig 2). When that is the case, the product $\tau(l_B - l_A)$ fully determines consistency, since for $x^*$ close to 1, the total rate of behavior switching is approximated by

$$\Omega(1, l_A) = \tau(l_B - l_A). \tag{6}$$

Thus, with a relatively weak sensitivity to the environment (i.e. $\tau$ low) and a small environmental impact differential (i.e. $l_A$ close to $l_B$), costly active behaviors can spread from low to high frequency provided the social pressure for active behavior is strong enough (Fig 2A–2C), and consistent adoption is expected (Fig 3A). In contrast, if the individual sensitivity to the environment is strong (large $\tau$) and/or the environmental impact differential is large (large $l_B - l_A$), behavioral inconsistency is expected, with agents frequently switching between active and baseline behaviors (Fig 3B and 3C), even if the predicted equilibrium frequency of the active behavior is high (see lower (yellow) regions in Fig 2D–2I, where the social pressure for $A$, $\delta_A$, is high hence the social normal threshold, $v$, is low).

For intermediate values of $\tau(l_B - l_A)$, low environmental reactivity, $\ell$, may be an additional cause of behavior inconsistency. In this case, decreasing environmental reactivity can destabilize the unique positive equilibrium and turn it into a limit cycle (Fig 2D–2F). Individuals will thus switch behavior at a total rate that is itself changing over a slower timescale set by the environmental reactivity. The slow timescale of environmental reactivity creates a time lag between the perceived environmental state and individuals' behavior, generating periodic oscillations in the switching rates.

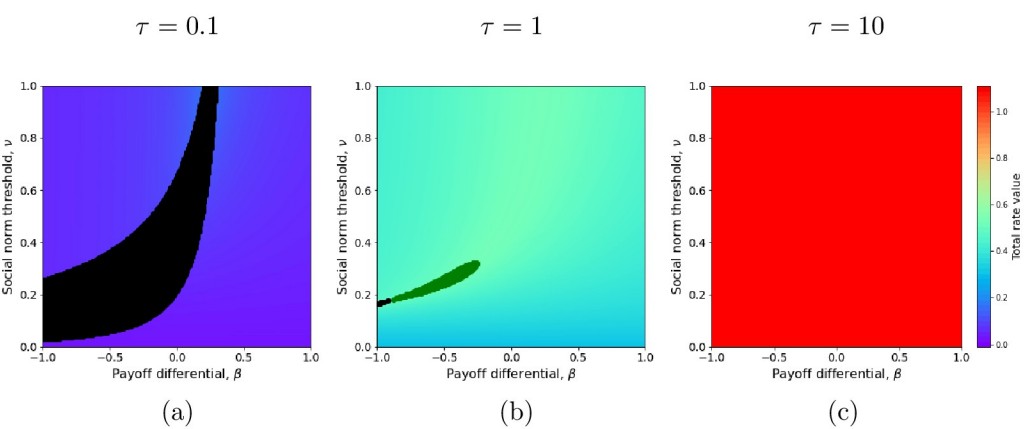

**Fig 3. Total switching rate at stationary state.** The total switching rate is equalt to the variance of the asymptotic fluctuations around the equilibrium $x^*$ as given by Eq 18. (a) Variance for $\tau = 0.1$. (b) Variance for $\tau = 1$. (c) Variance for $\tau = 10$. The environmental impact differential is fixed ($l_B = 1$, $l_A = 0.7$), environmental sensitivity is $\ell = 0.1$ and other parameters are set to their default values (Table 1).

## Robust environmental impact reduction through incremental behavior change

Starting from an environment perceived as strongly degraded, can an active behavior with much smaller environmental impact than the baseline behavior (i.e. large $l_B - l_A$) drive a robust improvement of the environmental state (from high to low $e$ at stationary state)? Such an active behavior could rise to high frequency provided the social pressure for proenvironmentalism is very strong (Fig 4A), with a concommitent drop in environmental impact (Fig 4D). But stochastic fluctuations in this case are large (Fig 4G), indicating that the active behavior may not be adopted consistently.

In contrast, environmental impacts and feedback such that $\tau(l_B - l_A)$ is small allow for the unconditional spread and consistent adoption of low-incentivized or even costly active behavior (i.e., $\beta$ close to 0 or even negative) (Fig 4B and 4H). Even though the effect of such behavioral change on the environmental state is small (Fig 4E), this sets the stage for a scenario of incremental behavioral change towards robust environmental impact reduction (Fig 4B and 4H) whereby increasingly more active behaviors would spread and be adopted consistently, each behavioral change imparting a small reduction of environmental impact. Once behavior $A$ is established, it becomes the common baseline behavior where individuals may start expressing a new active behavior $A'$, with lower environmental impact, and potentially a larger cost. In the latter case, a stronger social pressure (higher $\delta_A$ hence lower $v$) may compensate for the larger cost and ensure that the active behavior $A'$ spreads and becomes adopted consistently, in lieu of $A$. Thus, in a system where social conformism for active behavior can increase (increasing $\delta_A$ hence decreasing $v$) in relation with more effective active behavior (lower $l_A$) and/or the perception of reduced environmental impact (lower $e^*$), a substitution sequence of gradually more active (lower $l_A$) and more costly (more negative $\beta$) behaviors can take place, driving a potentially substantial decrease in the perceived environmental impact ($e^*$ decreasing to arbitrarily low levels).

We hypothesize that such an incremental scenario might be triggered even for active behaviors that are initially costly, and with weak social pressure for them (small $\delta_A$ hence high $v$)—as long as their cost is not too large so that bistability remains possible (upper areas of black regions in Fig 4B). In this case, random fluctuations in behavior frequency may cause the

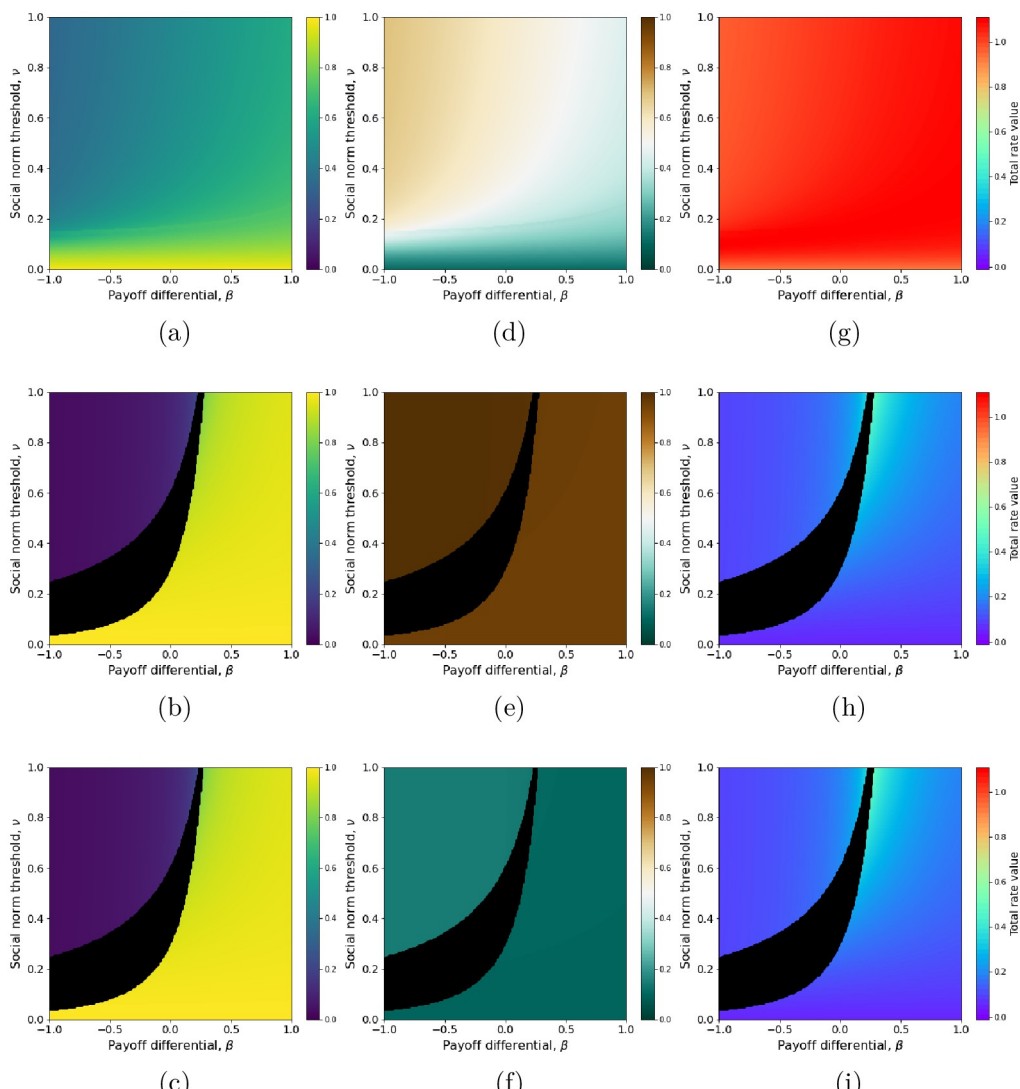

**Fig 4. Influence of the environmental impact differential, $l_B − l_A$, on the frequency of active behavior (a, b, c), perceived environmental state (d, e, f), and total switching rate (g, h, i) at equilibrium.** For (a), (d) and (g), the parameters are $l_B = 1$ and $l_A = 0.1$. For (b), (e) and (h), $l_B = 1$ and $l_A = 0.95$. For (c), (f) and (i), $l_B = 0.15$ and $l_A = 0.1$. Individual sensitivity to the environment and environmental reactivity are set to $\tau = 1$ and $\ell = 0.1$. Other parameters are set to their default values (Table 1).

system to switch from the low-frequency to the high-frequency $A$ equilibrium (Fig 5), where it may reside long enough for some new behavior $A'$ more active than $A$ to spread and take over. The study of such dynamics, involving the effect of individual stochasticity on alternate equilibria, is beyond the scope of this model and warrants further mathematical investigation.

## Discussion

We developed a simple mathematical model to study how social and environmental feedbacks jointly influence the consistent adoption of proenvironmental behavior. The treatment of individual assessment of the perceived environment and social interactions as two separate factors

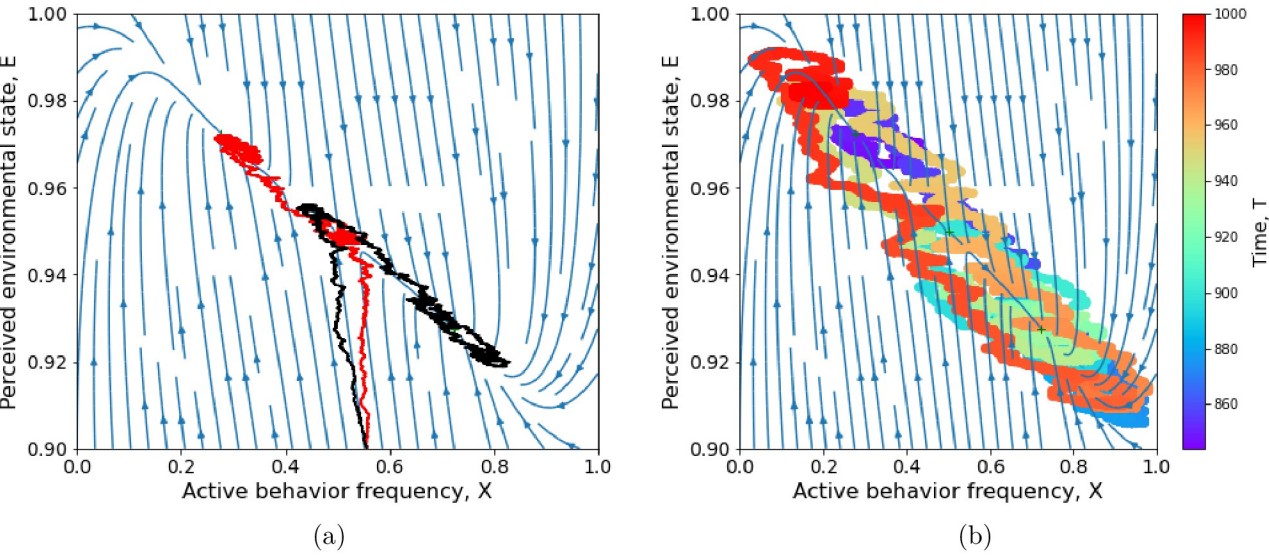

**Fig 5. Effect of stochastic fluctuations on behavior-environment dynamics in the bistable case.** (a) Convergence of two trajectories issued from the same initial condition to alternate equilibria. (b) A single trajectory, with color coding for passing time, visits alternate equilibria, from the higher $x^*$ (blue tones) to the lower $x^*$ (green) to the higher $x^*$ (orange) to the lower $x^*$ (red). The stochastic simulation algorithm is described in the Methods. Environmental sensitivity $\ell = 0.1$ (as in Fig 2A), payoff difference $\beta = -0.25$ and social norm threshold $\nu = 0.3$. Other parameters are set to their default values (Table 1).

of individual decisions differentiate our model from evolutionary games with environmental feedbacks, in which the influence of the environment is mediated by the payoffs of the strategic interactions [10–14] and even possibly by the game type varying in response to the agents' actions [22, 23]. Here we assume that the payoffs are constant and that the perception of environmental degradation can influence an individual's behavioral choice independently of their interactions with others—a similar assumption is made in models of environmental behavioral choice based on experiential learning [24]. This structure allows us to separate the behavioral effect of individual assessment of the perceived environmental state from the effect of social interactions. Our model thus aligns with the conceptual framework of Schill et al. (2019) [21] for connecting environmental behavior with both social and perceived biophysical contexts.

## The importance of timescales for behavioral consistency

The assumed invariance of the payoff difference (unaffected by variation in the environmental state variable) is rooted in the short timescale involved in the behavior-environment dynamics. Because the environmental state variable captures perceived information, shaped by behavioral intentions or actions, it is plausible that the timescale of environment-behavior change be short relative to the timescale over which the true state of the physical environment changes. Under this timescale separation, the payoff difference remains constant as the perceived environmental information changes, without precluding change in payoffs that could occur over the slower timescale of a changing physical environment.

When the timescale of individual assessment is fast relative to social interactions, the environmental feedback dominates the system dynamics, leading to inconsistent behavior. The relatively fast individual assessment timescale may originate from individuals having more confidence in their own evaluation of costs and benefits than in others' influence. This is known to occur, for instance, when the decision to be made carries a lot of personal weight

[25, 26] or when individuals have grown up in a very favorable environment [27]. When individual assessment is slow compared to social interactions, the social feedback dominates. This raises the question, whether, in practice, social influence could be stronger among individuals who engage in proenvironmental behavior than among individuals who do not. One can speculate that this could be the case if the active behavior is individually costly and perceived as a moral duty. In this case, the active individuals behave as cooperators whose efforts (measured in terms of opportunity cost) are influenced the most by the observation of the others' efforts [28, 29].

Overall, the timescale of perceived environmental change has little effect on the behavior-environment dynamics. Thus, whether individuals assume that their actions are environmentally meaningful in the short term (high environmental reactivity, $\ell$) or the long term (low environmental reactivity, $\ell$) generally has no significant effect on behavior consistency. The case of social interactions and individual assessment occurring on similar timescales is special, however. In this case, low environmental reactivity, $\ell$, creates a time lag between behavioral and perceived environmental changes, causing behavior-environment cycles when the proenvironmental behavior is costly and levels of conformism are not too different between behaviors. A similar effect of slow environmental reactivity relative to social interactions promoting oscillations was also detected by [30] in their model of forest growth and conservation opinion dynamics. Contrasted environmental impacts of behaviors $A$ and $B$ (i.e. large $l_B - l_A$) favor the limit cycle regime over bistability which is reminiscent of previous findings of behavior-environment cycles replacing bistability when the human influence on the environment is strong [31].

## Consistent adoption of incrementally more active behavior

A question of interest is how the consistent adoption of an active behavior depends on the magnitude of environmental impact reduction associated with that behavior. The model shows that for active behaviors causing only a small environmental impact reduction, the bistable regime is favored, which leads to consistent behavior adoption (of $A$ or $B$). In fact, a small environmental impact reduction by the active behavior has the same effect on the system dynamics as a slow timescale of individual assessment. Once a 'small-step' active behavior is adopted consistently, the perceived level of environmental degradation is only decreased by a small amount; but if more behavioral options were available, the socio-environmental context would be set to promote individuals engaging consistently in 'the next small step'. If the process were repeated, leading to the consistent adoption of active behaviors of gradually smaller environmental impact, we would expect the perceived level of environmental degradation to decrease. Interestingly, this might happen even if the relative cost of active behavior was increasing, provided the social pressure for active behavior increased concomitantly.

Our consideration of gradual behavioral change through a sequence of 'small steps' raises the empirical question of whether the perceived change in environmental state could in turn affect the repertoire of individual behaviors, and in particular motivate behaviors more active than $A$. In practice, the existence and direction of such an additional feedback may depend on whether each small step is individually beneficial and thus considered by people as a self-serving decision, or individually costly and considered as a form of cooperation. In the first case, there is no obvious reason for the perceived change in environmental state to affect individual decisions, so it is unlikely that such feedback would exist. In the second case, however, the question relates to the rich empirical literature on the influence of the perceived environmental state on cooperation. The majority of studies in this field, and in particular the highest powered studies, report a positive relationship between the quality of the environment experienced

by individuals and their level of investment in cooperation [32–39]; although some studies report opposite effects [40–42] or no effect at all [43, 44]. We may thus hypothesize the existence of the additional positive feedback whereby the perception of an improved environmental state would enlarge the behavioral repertoire and motivate more active behaviors. The improvement or, on the contrary, the deterioration of the perceived environment could lead individuals to invest more, or, on the contrary, less in proenvironmental behavior, thus generating the kind of behavioral sequence that we envisioned in this analysis.

## Limits and perspectives

Our work builds on the fundamental distinction between the individual's stable characteristics and the subset of situational characteristics which capture the social and environmental situatedness of behavior [45]. In the model, all parameters, except the rate of environmental reactivity, $\ell$, are set as individual characteristics. A key assumption is that all individuals are identical in their stable characteristics. Our framework could be extended to relax this assumption and investigate the consequences of heterogeneity in individual social status or personalities [46, 47]. For example, the same objective cost of the active behavior (e.g. buying or maintaining an electric car) may be perceived very differently depending on the individual's wealth [48, 49]. Likewise, individuals of different social status may vary in their experience of social pressure from individuals expressing the active vs. baseline behavior; this in the model would manifest through inter-individual variation of $\delta_A$ and $\delta_B$ [27].

Given the predicted importance of the individual sensitivity to the environment, $\tau$, and environmental impact differential, $l_B - l_A$, the outcome (consistent adoption of the active behavior) is likely to be influenced by inter-individual heterogeneity in these two parameters. It is known that individuals can differ greatly in their perception and assessment of the state of degradation of their environment, due to differences in social origin, education, or information [49, 50]; and in their potential proenvironmental response to perceived environmental degradation [50]. This heterogeneity could result in wide variation of both $\tau$ and $l_B - l_A$ among individuals, with contrasted personalities such as being little responsive and acting weakly (small $\tau$ and $l_B - l_A$), or responding fast and strongly (large $\tau$ and $l_B - l_A$).

In previous models of coupled human behavior and natural environment dynamics, the environment typically is measured by a variable such as the abundance of a renewable resource (e.g. forests [17], fisheries [51]), or a physical variable such as atmospheric greenhouse gases concentration or temperature [52]. In these models, the environmental dynamics are driven by their own endogenous processes and impacted by human activities (gas emissions, harvesting. . .). These models ask how human behavioral feedbacks alter the stability properties of the perturbed (polluted, exploited. . .) ecosystem. An important difference between our approach and previous human-environment system models lies in our definition of the environmental state in terms of perceived degradation or vulnerability, rather than actual physical components (atmospheric $CO_2$ level, abundance of a natural resource. . .) of the environment. This information changes under the influence of individuals' intentions or behaviors; the physical environment (or physical environmental trend, as with global climate warming or biodiversity loss) may also change as a consequence, but we assume that the change would occur on a much slower timescale and therefore has no influence on the individual decisions that the model describes. For example, in the case of global climate change, our approach assumes that our world is already locked into warming: even if we stopped emitting greenhouse gases today, it would take several decades before we observe curbing in the rise of global temperature. None-the-less, rapid change in individual behavior may occur, including the spread of pro-

climate action or intention, concomitantly with the perception that the level of environmental degradation or vulnerability decreases.

This representation of the environment allowed us to make the assumption that the payoff difference is insensitive to the environmental state variable. Note, however, that a feedback of the environment to the payoff difference can be seen as implicit in our scenario for the consistent adoption of incrementally more active behavior. This could be due to institutional intervention responding—on a slow timescale—to change in the environmental stationary state, as defined here, and altering the payoff difference through incentives (reducing the payoff difference) or by promoting more effective behaviors that would also be more costly (increasing the payoff difference). Another significant difference with most previous human-environment models is that here the environment, being purely informational, does not have its own intrinsic dynamics. A similar assumption was made by Weitz et al. (2016) [12] in a game-theoretic model where the environmental feedback was positive and governed by a tipping point. Future work could address the effect of a positive feedback in our model, to capture positive reinforcement (improved environmental state encourages to do more [49, 53]) and "giving up" (an environment assessed as degraded leads to less effort, rather than more [49, 53]).

## Concluding remarks

In this minimal model of behavior-environment feedback, individuals intend proenvironmental action when they perceive their environment as strongly degraded or vulnerable, and relax their effort as they perceive amelioration of the environmental state. This negative environmental feedback, by itself, opposes the consistent adoption of costly proenvironmental behavior, but opens the possibility for such behavior to rise from very low frequency and reach a tipping point at which social interactions and conformism will pick up and drive its consistent adoption. This suggests a scenario to achieve large reductions in environmental impact, which involves the consistent adoption of incrementally more active and potentially more costly behavior. The theoretical conditions for this to occur (social interactions timescale faster than individual assessment timescale, strong social pressure for proenvironmental action) raise empirical questions regarding why individuals' environmental intentions would rely more on others' influence than on their own assessment of the environmental state, and what factors (individual or collective) can make the social pressure of conformism stronger for proenvironmentalism.

## Methods

In a population of finite size $N$, at any time $t$, the perceived environmental state and the numbers of individuals who are performing $A$ or $B$ are denoted by $E_t^N$, $N_t^{A,N}$ and $N_t^{B,N}$, respectively. Since the population size is constant we have $N_t^{B,N} = N - N_t^{A,N}$. Hereafter we derive a stochastic model for the joint dynamics of the frequency of behavior $A$ in the population, $X_t^N = \frac{N_t^{A,N}}{N}$, and the perceived environmental state, $E_t^N$. The list of parameters and their default values is given in Table 1.

### Environment dynamics

We assume that the dynamics of the perceived environnemental state $E_t^N$ follows a deterministic continuous process. Each individual in the population has the same perception of the environment. The dynamics of $E_t^N$ is driven by the ordinary differential equation

$$\dot{E}_t^N = h(X_t^N, E_t^N),$$

where $h$ captures the environmental impact of the two behaviors given their frequency, according to

$$h(x, e) = \ell e (l_A x + l_B (1 - x) - e) \tag{7}$$

for any $(x, e)$ in $[0, 1] \times [l_A, l_B]$.

The function $h$ is chosen such that in a population where all individuals express behavior $A$ ($B$, respectively), the rate of change of the environment perceived as minimally (maximally) degraded is proportional to $l_A$ ($l_B$) and the stationary value of the perceived environmental state is $l_A$ ($l_B$).

## Behavior dynamics

Two factors influence the decision process of modeled agents: frequency of encounters with other modeled agents, and the value of the state variable representing some environmental factor.

**Social interactions.** We further assume

$$\begin{aligned} g_A(X^N) &= X^N \\ g_B(X^N) &= 1 - X^N. \end{aligned} \tag{8}$$

An agent with behavior $i$ switches to behavior $j$ via social interactions at rate

$$\lambda_{i \to j}^N(X^N) = N^2 \kappa g_i(X^N) g_j(X^N) \lambda_j^N(X^N), \tag{9}$$

where $\lambda_i^N(x)$ is the individual attractiveness of behavior $i$, $N^2 \kappa g_i(X^N) g_j(X^N)$ is the number of potential encounters. Note that the number of potential encounters is symmetrical, $g_i(X^N) g_j(X^N) = X^N(1 - X^N)$. The difference between behavior $i$'s and $j$'s rate rely on the individual attractiveness of behavior. The individual attractiveness of behavior $i$ is taken of the form

$$\lambda_i^N(X^N) = \frac{1}{N} (\gamma_i + \delta_i g_i(X^N)). \tag{10}$$

As a result, the individual rate of behavioral switch from $i$ to $j$ is

$$\lambda_{i \to j}^N(X^N) = N \kappa X^N (1 - X^N)(\gamma_j + \delta_j g_j(X^N)). \tag{11}$$

The term $\delta_j g_j(X^N)$ reflects that social influence is a coercive mechanism which encourages conformism.

**Individual assessment.** Behavior switching based on the assessment of the state of the environment occurs at the individual rate

$$\tau_{i \to j}^N(X^N, E^N) = N g_i(X^N) \tau_j(E^N), \tag{12}$$

where $g_A(X^N) = X^N$ and $g_B(X^N) = 1 - X^N$ (as above) and the environmental perception of individual $i$ is measured by $\tau_j(E^N)$, which captures the difference between the environmental state, $e$, and the contribution of behavior $j$ to the degradation of the environmental state.

The simplest form then is

$$\begin{aligned} \tau_A(E^N) &= \tau(E^N - l_A) \\ \tau_B(E^N) &= \tau(l_B - E^N). \end{aligned} \tag{13}$$

Thus, the larger the difference between the perceived environmental state, $E^N$, and the individual environmental impact of behavior $i$, $l_i$, the faster an agent expressing $j$ behavior switches to $i$ behavior.

## Dynamics of the behavior-environment stochastic process

The dynamics of the coupled behavior-environment process $(X_t^N, E_t^N)$ are stochastic, driven by the probabilistic events of agents switching between the baseline ($B$) and active ($A$) behaviors, under the joint effects of encounters and individual assessment, and the deterministic response of the perceived environmental state. Mathematically, the effects of all possible events (agents' behavior switches, change in perceived environment) on the state of the Markovian system $(X_t^N, E_t^N)_{t \geq 0}$ are captured by the infinitesimal generator $L^N$ of the stochastic process $(X_t^N, E_t^N)$. For $(x, e) \in [\![0, \frac{1}{N}, \cdots, 1]\!] \times \mathbb{R}_*^+$ and a test function $f \in \mathcal{C}_b^1([\![0, \frac{1}{N}, \cdots, 1]\!] \times \mathbb{R}_*^+, \mathbb{R})$, we have

$$
\begin{aligned}
L^N f(x, e) = \quad & N^2 \kappa x(1-x) \lambda_A^N(x) \left[ f\left(x + \frac{1}{N}, e\right) - f(x, e) \right] \\
& + N^2 \kappa x(1-x) \lambda_B^N(x) \left[ f\left(x - \frac{1}{N}, e\right) - f(x, e) \right] \\
& + N(1-x) \tau_{B \to A}(x, e) \left[ f\left(x + \frac{1}{N}, e\right) - f(x, e) \right] \\
& + N x \tau_{A \to B}(x, e) \left[ f\left(x - \frac{1}{N}, e\right) - f(x, e) \right] \\
& + h(x, e) \frac{\partial f(x, e)}{\partial e}.
\end{aligned}
\tag{14}
$$

Agents switch behavior at a given time $t$ for a given state of the system $(X_t, E_t)$ with a probability given by Eq (14). In this expression, the first and second rows account for individual behavior switches due to encounters (from $B$ to $A$ or $A$ to $B$, respectively). For instance, the rate at which a $B \to A$ switch occurs because of encounters (first row) is proportional to $N(1 - x)$, the number of agents adopting behavior $B$; $\kappa N x$, the encounter rate between a single agent adopting $B$ and agents adopting $A$; and $\lambda_A(x)$, the social attractiveness of a single agent adopting $A$. The third and fourth rows account for switches because of individual assessment of the perceived environment state. For instance, the rate at which a $B \to A$ switch occurs because of the environmental state variable (third row) is proportional to $N(1 - x)$, the number of agents adopting $B$; and $\tau_{B \to A}(x, e)$, the rate at which an agent in state $B$ adopts the alternate behavior $A$ after assessing the impact of its behavior on the perceived state of the environment. Finally, the last row accounts for changes in the perceived environmental state depending on the frequency. By taking $f(x, e) = x$ and $f(x, e) = e$, we obtain the deterministic part in Eq. (S1.2) in S1 Appendix [54] while $f(x, e) = x^2$ gives the quadratic variation in Eq. (S1.4) in S1 Appendix. The process whose law is characterized by Eq (14) is a Piecewise Deterministic Markov Process where the population state (frequencies of behaviors) probabilistically jumps at each change in agent behavior while the environmental state deterministically and continuously changes between jumps.

## Dynamical system approximation for large populations

In Section 2 in S1 Appendix [54], we explain that, assuming the population size $N$ very large and under Assumptions (10–15), the sequence of stochastic processes $(X^N, E^N)_{N \in \mathbb{N}^*}$ converges

in distribution to the unique solution of the system Eq (1). By plugging Eqs (2) and (3) into Eq (1), we obtain Eq (15)

$$\frac{dx_t}{dt} = p(x_t, e_t) = \kappa x_t (1 - x_t)[\beta + \delta_A x_t - \delta_B (1 - x_t)] + \tau[e_t - l_A(1 - x_t) - l_B x_t]$$

$$\frac{de_t}{dt} = h(x_t, e_t) = \ell e_t (l_A x_t + l_B (1 - x_t) - e_t) \tag{15}$$

with initial conditions $x_0$ between 0 and 1 and $e_0$ between $l_A$ and $l_B$. The first equation governs the frequency $x$ of the active behavior, $A$. The second equation drives the dynamics of the perceived environmental state variable, $e$.

The model predicts up to three equilibria given by the zeros of

$$p(x, l_A x + l_B(1 - x)) = p_0(x) + \tau(l_B - l_A)(1 - 2x), \tag{16}$$

($p_0$ defined in Eq (6)), that are nonnegative and less than (or equal to) one. By setting both $\tau > 0$ and $\kappa > 0$ in Eq (15), the effect of the environmental feedback on its own can be highlighted by comparing Eq (15) at its stable equilibria with the value of Eq (16) at the same state (i.e. $x^* = 0$ or $x^* = 1$). Since $p(0, l_B) = \tau(l_B - l_A) > 0$ and $p(1, l_A) = -p(0, l_B) < 0$, the environmental feedback moves the equilibria of the system away from 0 and 1 (Fig 2). The roots of Eq (16) also show that the number of equilibria is likely influenced by the encounter rate, $\kappa$, the payoff differential, $\beta$, the social norm threshold, $v = \frac{\delta_B}{\delta_A + \delta_B}$, and the combination (product) of the individual sensitivity to the environment, $\tau$, and the differential environmental impact, $l_B - l_A$.

## Quantifying the effect of individual stochasticity

To analyse the fluctuations of the stochastic model around the deterministic limit, we generalize the central limit theorem to the convergence of the stochastic process $(X_t^N, E_t^N)_{t \in [0,T]}$ to the deterministic solution of Eq (1). We therefore introduce
$(\eta_t^N)_{t \in [0,T]} = (\eta_t^{A,N}, \eta_t^{E,N})_{t \in [0,T]} = (\sqrt{N}(X_t^N - x_t, E_t^N - e_t))_{t \in [0,T]}$, where $((x_t, e_t), t \le T)$ is the deterministic solution of Eq (1) and $(X^N, E^N)$ is the stochastic process. Assuming that $\eta_0^N$ converges in law to $\eta_0$, when $N \to \infty$, the process $(\eta_t^N)_{t \in [0,T]}$ converges in law to a Ornstein-Uhlenbeck type process $(\eta_t)_{t \in [0,T]} = (\eta_t^A, \eta_t^E)_{t \in [0,T]}$ and we have

$$(X_t^N, E_t^N) = (x_t, e_t) + \frac{1}{\sqrt{N}}(\eta_t^A, \eta_t^E) + o\left(\frac{1}{\sqrt{N}}\right).$$

For all $t$ in [0, T], the process $(\eta_t)_{t \in [0,T]} = (\eta_t^A, \eta_t^E)_{t \in [0,T]}$ satisfies

$$\eta_t^A = \eta_0^A + \int_0^t [(\kappa(1 - 2x_s)(\gamma_A - \gamma_B - \delta_B + (\delta_A + \delta_B)x_s) - \tau(l_B - l_A))\eta_s^A + \tau \eta_s^E]ds$$

$$+ \int_0^t \sigma(x_s, e_s)dW_s, \tag{17}$$

$$\eta_t^E = \eta_0^E + \int_0^t \ell[(l_A - l_B)e_s \eta_s^A + (l_A x_s + l_B(1 - x_s) - 2e_s)\eta_s^E]ds.$$

where

$$\sigma(x, e) = \sqrt{\kappa x(1 - x)(\lambda_A(x) + \lambda_B(x)) + \tau_A(e)(1 - x) + \tau_B(e)x}, \tag{18}$$

and $W$ is a standard Brownian motion (see Section 4 in S1 Appendix [54] for mathematical detail).

Note that the drift and variance are functions of the solution of the deterministic system (1). The stochastic part of the fluctuation process given by Eq (18) is the square root of the total rate $\Omega$ (see Eq (4)) of behavior switching in the population. According to Eq (17), it only affects the fluctuations in behavior frequency (not in the perceived environmental state).

## Simulations and numerical analysis

For the stochastic process, the dynamics of agent behaviors' frequency (by stochastic jumps) is jointly simulated with the dynamics of the perceived environment (by deterministic changes, continuously in time between the stochastic population jumps). Random times are for any $N$ drawn according to an exponential distribution of parameters $\xi^N$, where

$$\xi^N > N \sup_{(x,e)\in[0,1]\times[l_A,l_B]} (\kappa(\lambda_A(x) + \lambda_B(x)) + \tau_A(e) + \tau_B(e)). \tag{19}$$

At each of these times, we update our variables of interest. There are three possible cases: either no agent changes their behavior in the population, or one agent switches from $B$ to $A$, or one agent switches from $A$ to $B$. The perceived environment is changed using a Euler scheme between two events in the population.

Without loss of generality, parameters $\kappa$ and $l_B$ are fixed to 1 (default values for parameters used in numerical analyses are reviewed in Table 1). We analyse the properties of the stochastic and deterministic models for values of $\delta_A$ and $\gamma_A$ spanning the whole range of possible values while keeping $\delta_B$ and $\gamma_B$ constant. Parameters are varied across a discrete grid. We search for fixed points by computing the zeros of the polynomial given by Eq (16). Local stability is tested by computing the Jacobian matrix of the system. We use the Poincaré-Bendixson theorem to check the absence of limit cycle (Th.1.8.1 in [55], see also Section 3.2 in S1 Appendix [54]). When the existence of a stable limit cycle in addition to an attractive fixed point cannot be excluded, we simulate the dynamical system for different initial conditions. Would there be a limit cycle crossing the trajectory of the simulations, the trajectory would be trapped around the limit cycle and not converge to its stable fixed point. Otherwise, all trajectories converge to the equilibrium, thus excluding the existence of a limit cycle.

## Supporting information

**S1 Appendix. Supplementary.** Mathematical explanations.
(PDF)

**S1 File. Read me file.** Explanations on how to use the S1 Code.
(TXT)

**S1 Code. Code used to generate images for this article and S1 Appendix.**
(PY)

## Acknowledgments

We thank Sylvie Démurger, Jean-Stéphane Dhersion and the team of the *Mission pour les Initiatives Transverses et Interdisciplinaires* at the French National Center for Scientific Research (CNRS) for the initial impetus and support to develop this research program. We are grateful for the stimulating work environment that the summer school organized by the Chair "Modélisation Mathématique et Biodiversité" has provided.

## Author Contributions

**Conceptualization:** Claire Ecotière, Sylvain Billiard, Régis Ferrière, Sylvie Méléard.

**Formal analysis:** Claire Ecotière.

**Funding acquisition:** Régis Ferrière.

**Investigation:** Claire Ecotière, Sylvain Billiard, Jean-Baptiste André, Pierre Collet, Régis Ferrière, Sylvie Méléard.

**Methodology:** Pierre Collet.

**Project administration:** Régis Ferrière, Sylvie Méléard.

**Supervision:** Sylvain Billiard, Régis Ferrière, Sylvie Méléard.

**Writing – original draft:** Claire Ecotière, Sylvain Billiard, Régis Ferrière, Sylvie Méléard.

**Writing – review & editing:** Claire Ecotière, Sylvain Billiard, Jean-Baptiste André, Pierre Collet, Régis Ferrière, Sylvie Méléard.

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
