## [Decision Letter · Decision Letter 0]

7 Mar 2023

Dear Phd Student Ecotiere,

Thank you very much for submitting your manuscript "Human-environment feedback and the consistency of proenvironmental behavior" for consideration at PLOS Computational Biology.

As with all papers reviewed by the journal, your manuscript was reviewed by members of the editorial board and by several independent reviewers. In light of the reviews (below this email), we would like to invite the resubmission of a significantly-revised version that takes into account the reviewers' comments.

The manuscript has been evaluated by three expert reviewers. They all find the results potentially interesting, but two of them also raise some serious concerns. For example, reviewer #1 raises some excellent questions about some of the model assumptions and the main motivation for this study. Reviewer #3 suggests to describe the model and the model results more carefully. I would like to encourage the authors to carefully address these concerns when revising their manuscript. In addition, please also provide the code used to derive the results (ideally the code is uploaded to some repository, with a readme file that explains how to use the code).

We cannot make any decision about publication until we have seen the revised manuscript and your response to the reviewers' comments. Your revised manuscript is also likely to be sent to reviewers for further evaluation.

Sincerely,

Christian Hilbe

Academic Editor

PLOS Computational Biology

James O'Dwyer

Section Editor

PLOS Computational Biology

The manuscript has been evaluated by three expert reviewers. They all find the results potentially interesting, but two of them also raise some serious concerns.

For example, reviewer #1 raises some excellent questions about some of the model assumptions and the main motivation for this study. Reviewer #3 suggests to describe the model and the model results more carefully.

I would like to encourage the authors to carefully address these concerns when revising their manuscript.

In addition, please also provide the code used to derive the results (ideally the code is uploaded to some repository, with a readme file that explains how to use the code).

Reviewer's Responses to Questions

**Comments to the Authors:**

Reviewer #1: #### Summary

In this manuscript, the authors engage in the question of how pro-environmental behavioral consistency is affected by social norms and the individual assessment of the environmental state and how, in turn, behavior affects the perceived environmental state. To answer this question, they develop an individual-based model which captures behavior change and environmental perception via transition rates or transition timescales. The majority of the analysis is performed via a set of deterministic differential equations, which approximate the individual-based stochastic model and resemble imitation/replicator dynamics from evolutionary game theory. Behavioral consistency is operationalized by the population equilibrating to either fully adopting option B (baseline business-as-usual impact) or option A (active pro-environmentalism). The authors show that the interaction of social norms and individual environmental assessment is more important for behavioral consistency than the timescale of how the perception of the environment changes.

#### Overall assessment

I appreciate how the authors motivate their model, coming from a relevant question in the sustainability discourse. The manuscript is clearly written. The model is derived and presented with sufficient mathematical detail and analyzed intensively. However, I believe, quite some improvements can be made. More work is needed to communicate the originality and innovation of this work (Comments 1-2), elaborate on the model design and analysis (Comments 3-4), and discuss how the model links to reality (Comments 5-6).

#### Comments

1. I still struggle to see what the main contribution of this work is. Is it a methodological advancement for human-environment modeling? Or a contribution to the question of why people behave inconsistently in their pro-environmental behavior? I believe the authors need to elaborate more on this point and critically discuss the main contribution of their work.

2. I found only 5 references to other human-environment models the authors use to discuss and compare their work against. A more thorough comparison with the literature on human-environment modeling is needed. Especially works on evolutionary game theory and individual learning dynamics with environmental feedback are closely related to the authors' modeling approach. Embedding their model in this literature would help to better contextualize and relate it.

3. Some model design choices remain opaque to me. For example, why does the environmental state have no influence on the payoff difference? The authors make the case that in their model, the environmental state only presents perceived information rather than the true state of a physical environment. But if payoff feedback from the environment is excluded from the model, what does that mean for the validity of the timescales considered? It is not plausible to me to assume that this feedback plays only a negligible role for medium or long time scales. I believe, this design choice deserves more attention during the derivation of the model.

4. I appreciate the fluctuation analysis the authors perform from a mathematical point of view. But what can we learn from it?

5. The authors state "Environmental feedback tends to generate behavioral inconsistency, which eventually limits the environmental impact reduction of active behavior". I wonder whether the second part of this sentence is true. I can see that including environmental feedback in the behavioral adaptation processes in addition to the social norm component causes more mixed strategy equilibria (Fig 2, comparing tau=0.1 with tau=1.0). This means that previously fully active parameter regions are less active now. But this also means that previously full non-active parameter regions have become somewhat active. There, it may be better to be inconsistent with your pro-environmental behavior than to show no pro-environmental behavior at all.

6. The authors' model interpretation regarding the question of behavioral consistency is as follows: "If x converges to 1, individuals perform behavior A most of the time, which means that the individual switching rate from A to B vanishes. In other words, behavior A is expressed consistently. If x converges to 0, the individual switching rate from B to A vanishes and behavior B is expressed consistently. In all other cases, individual behavior is inconsistent." Let me challenge this interpretation. Imagine a standard (one-population) imitation/replicator dynamics in the chicken/hawk-dove/snowdrift game. The dynamics converge to a mixed strategy, but does this represent behavioral inconsistency? Isn't it likewise plausible to assume that a fraction of the population consistently plays one strategy and the rest of the population plays the other?

#### Minor comments

* The references to the figures from the main text can be improved. For example, the first time the reader sees a reference to Figure 1 is when they are encouraged to compare Figure 1 with Figure 2.

* When introducing a new model it can only help to have descriptive parameter names, as the authors have. The authors could improve their presentation of the figures and the text by naming the descriptive parameter names in addition to the corresponding symbol.

Reviewer #2: This paper addresses some current issues relevant to everyone's life. By considering a coupled system for the environment and the social public goods game, the authors explore the model through theoretical and numerical analysis, trying to reach meaningful conclusions to guide policymaking and the general public. It is good for publication in my opinion. Here are some questions and comments on minor issues.

Page 2 Abstract line 11: "little sensitive", maybe change to "indifferent"

Abstract line 15: "Noticingly", should be "Noticeably"

Page 3 Introduction line 5: remove "to make"

Page 4 line 2: "change dynamically with and feed back continuously", change to "change dynamically with continuous feedback"

Page 4 Results Model overview line 2: the variable E and later the symbol e seem to both represent the environmental state. Maybe the authors should try to pick one of them and be consistent with the notations.

Page 4 Results Model overview line 3: "with larger E meaning that the environment is perceived as more degraded", change to "with larger E as more degraded environment"

Page 5 lines 2 and 3: the A behavior, the B behavior, can be replaced by behavior A and behavior B

Page 6 line 2 from the bottom up: "was allowed to arise" change to "arose"

Page 7 line 5: I believe the fixed point x^* should be (\\delta_B-\\beta)/(\\delta_A+\\delta_B)

Page 7 Effect of environmental feedback on behavior dynamics Equation (4): What is p(x) here? What is p_0(x) here? Are they supposed to be the right-hand side of the first equation in equation (2)? Maybe add that e is always positive, so the only option for equilibrium is e=l_A x+l_B(1-x)

Page 8 line 2: Figure 1 and 2. I'm not sure what the format requirement of the journal is, but it would be very helpful if the figures mentioned are actually presented on the same page here

Page 8 Conditions for spread and consistency of active behavior line 3: "can be propagated and stabilized" change to "can propagate and stabilize"

Page 8 Conditions for spread and consistency of active behavior line 6: "and adopted" change to "and be adopted"

Page 9 line 8: rewrite "drive the rise of A frequency close to 1"

Page 9 line 14: add "l" after "environmental reactivity"

Page 9 line 2 and 4 from the bottom and the section title: "stationary" misspelled

Page 10 line 3: "behavioral" change to "behavior"

Page 10 paragraph 3 line 3: E^* has not been defined anywhere

Page 11 line 2: typo "further"

Page 14 bottom two lines: rewrite "We refer to ...Table 1"

Page 15 Social interactions equation (6): why is the right hand side only dependent on j? shouldn't it depend on both i and j just like in equation (10)? Also what would be the formula for i to i and j to j?

Page 16 lines under equation (10): rewrite "tau_A and tau_B must capture ... , compared to the alternative behavior". maybe try to explain what tau_A and tau_B represent first, and then describe what rule/law they should follow

Page 17 lines 2 and 3: Maybe add a little explanation to what f(x,e) represents.

Page 17 line above Quantifying the effect of individual stochasticity: Plug equations (14), (15) and (1) into (13), we get equation (2).

In the supplementary material, section 3.1, equation (8) has some minor errors, I think maybe the rescale r should just be l over tau, there is no l_B, and let c just be l_A/(l_B-l_A), then the right hand side of the second equation of (8) would be r(v+c)(1-x-v). Because of this, the Jacobian in section 3.2 also needs some modification.

Supplementary material Section 3.4 line 2: equation (18), I don't think there is an equation (18) in the supplementary material

Reviewer #3: This paper presents an extension of the family of "bioeconomic" models for natural resource management developed in the 1970's by Colin Clark (see Mathematical Bioeconics, 1976, 1990, 2005). The extension adds 'social interaction'(replicator dynamics) between two behavior types that determine environmental impact (harvest in bioeconomic models). The paper reports on the impact of various assumptions about model parameters on the characteristics of equilibria. As such, I struggle a bit with stretching such a simple model to describe 'Behavior dynamics' and 'social interactions'. Yes, the model describes the behavior of very, very simple algorithms, not humans. I think the paper should soften language. i.e. this line on page 15 (there are no line numbers so the page number is the best I can do.....)

"Two factors influence individual behavior: social interactions and individual assessment of the environmental state."

This sounds like we are talking about sentient beings. This is misleading. Something like

"Two factors influence the decision process of modeled agents: frequency of encounters with other modeled agents, and the value of the state variable representing some environmental factor."

is more accurate and gives a sense to the reader what the paper is actually about. It is not really about human-environment feedback and the consistency of proenvironmental behavior, but rather, a very simple model of interacting particles that captures some general features of human-environment feedback and the consistency of proenvironmental behavior. I think the title should include the world modeling, simulating, etc. to more accurately describe what the paper is about.

Along the same lines, the links between the language used to describe model behavior and the model itself is sometimes not clear. I searched 'behavioral consistency' in the PDF document. To the extend I can rely on the search results, it appeared 11 times, yet it is never explicitly defined. Of course, I can guess what it is, but it would be nice to be able to confirm my understanding with that of the authors.

Finally, the discussion is a bit long and technical with many suggestions for further research. Perhaps readers of this journal will appreciate this. For my part, given the simplicity of the model and rather intuitive nature of the results, I am missing a good punch line or some sort of interesting empirical puzzle that this analysis sheds new light on. Yes, this is the journal of computational biology, but some more interesting motivation for and insights from the computation would improve the paper substantially.

**Have the authors made all data and (if applicable) computational code underlying the findings in their manuscript fully available?**

Reviewer #1: **No: **I did not find computational code that was used to generate the figure or simulate the stochastic process. All equations and a description of the algorithm, however, are given.

Reviewer #2: **No: **The authors presented figures of their simulations but didn't provide the original code they used.

Reviewer #3: Yes

PLOS authors have the option to publish the peer review history of their article (what does this mean?). If published, this will include your full peer review and any attached files.

Reviewer #1: No

Reviewer #2: No

Reviewer #3: No
---

## [Decision Letter · Decision Letter 1]

19 Jun 2023

Dear Phd Student Ecotiere,

Thank you very much for submitting your manuscript "Human-environment feedback and the consistency of proenvironmental behavior" for consideration at PLOS Computational Biology. As with all papers reviewed by the journal, your manuscript was reviewed by members of the editorial board and by several independent reviewers. The reviewers appreciated the attention to an important topic. Based on the reviews, we are likely to accept this manuscript for publication, providing that you modify the manuscript according to the review recommendations.

Sincerely,

Christian Hilbe

Academic Editor

PLOS Computational Biology

James O'Dwyer

Section Editor

PLOS Computational Biology

The paper has been improved substantially, and it's quite likely that it will be accepted eventually.

Before that, however, I would like the authors to address the remaining points raised by reviewer #1.

Reviewer's Responses to Questions

**Comments to the Authors:**

Reviewer #1: The authors addressed all of my comments in the revised version of the manuscript. Now it becomes much more apparent that the model assumes the behavior is driven directly by environmental perception and not a payoff difference resulting from environmental changes.

Three remaining pointers concern the model's interpretation and validity.

- In the model, the perception of the environmental state is proportional to the pro-environmental active individuals. Why should the individual's private environmental experience depend on the behavior of others when considering the actual state of the environment as not changing? I can only think of societal movements or cultural trends, like climate protests, plant-based diets, or flight shaming, where such a dynamic seems plausible. But these are purely social processes without environmental feedback - precisely because the environment is not changing.

- The choice of modeling constant environments has consequences for the validity of the model's time range. Letting the social process run for longer than one assumes the environment to be constant makes the model inconsistent. For example, in the face of global change, we cannot consider the environment constant on time scales longer than decades. This would mean that all social processes in the model must operate on time scales much shorter than decades - let's say on a yearly time scale. What time interval does a model step represent? A day? How long does it take until equilibrium is reached? It would help greatly if the authors estimate the actual timescales (days, weeks, months, years, decades) on which their model operates.

- The authors distinguish their model from evolutionary games with environmental feedback in the feature of perceived environmental and social interactions as two separate factors of individual decisions - referencing Schill et al., who argue for connecting environmental behavior with both social and biophysical contexts. However, behavioral choice independent of interactions with others has also been operationalized with models of individual learning. See, for example, single-agent models:

- Lindkvist, E., & Norberg, J. (2014). Modeling experiential learning: The challenges posed by threshold dynamics for sustainable renewable resource management. _Ecological Economics_, _104_, 107-118.

- Lindkvist, E., Ekeberg, Ö., & Norberg, J. (2017). Strategies for sustainable management of renewable resources during environmental change. _Proceedings of the Royal Society B: Biological Sciences_, _284_(1850), 20162762.

- Strnad, F. M., Barfuss, W., Donges, J. F., & Heitzig, J. (2019). Deep reinforcement learning in World-Earth system models to discover sustainable management strategies. _Chaos: An Interdisciplinary Journal of Nonlinear Science_, _29_(12), 123122.

- Chapman, M., Scoville, C., Lapeyrolerie, M., & Boettiger, C. (2022). Power and accountability in reinforcement learning applications to environmental policy. _arXiv preprint arXiv:2205.10911_.

but also more recently in multi-agent situations:

- Barfuss, W., Donges, J. F., Vasconcelos, V. V., Kurths, J., & Levin, S. A. (2020). Caring for the future can turn tragedy into comedy for long-term collective action under risk of collapse. _Proceedings of the National Academy of Sciences_, _117_(23), 12915-12922.

- Huang, F., Cao, M., & Wang, L. (2020). Learning enables adaptation in cooperation for multi-player stochastic games. _Journal of the Royal Society Interface_, _17_(172), 20200639.

- Zhang, T., Williams, A., Phade, S., Srinivasa, S., Zhang, Y., Gupta, P., ... & Zheng, S. (2022). AI for Global Climate Cooperation: Modeling Global Climate Negotiations, Agreements, and Long-Term Cooperation in RICE-N. _arXiv preprint arXiv:2208.07004_.

Reviewer #2: The revision added explanations to parameters in the model and general conceptual understanding of the model. I think it is ready for publication after rewriting "Individuals perceive having an impact on the environment to a degree that is determined by their behavior" on page 4 lines 88-89.

Reviewer #3: My concerns have been adequately addressed.

**Have the authors made all data and (if applicable) computational code underlying the findings in their manuscript fully available?**

Reviewer #1: **No: **The PDF said so in the beginning, but I could not find an actual Data or Code Availability Statement and, therefore, was not able to look at the code.

Reviewer #2: Yes

Reviewer #3: None

PLOS authors have the option to publish the peer review history of their article (what does this mean?). If published, this will include your full peer review and any attached files.

Reviewer #1: No

Reviewer #2: No

Reviewer #3: No

Figure Files:

Data Requirements:

Reproducibility:

References:

---

## [Editor Report · Decision Letter 2]

13 Aug 2023

Dear Phd Student Ecotiere,

We are pleased to inform you that your manuscript 'Human-environment feedback and the consistency of proenvironmental behavior' has been provisionally accepted for publication in PLOS Computational Biology.

Best regards,

Christian Hilbe

Academic Editor

PLOS Computational Biology

James O'Dwyer

Section Editor

PLOS Computational Biology

After the previous two rounds of review, I asked the authors to incorporate the remaining comments of reviewer #1. The authors have taken this request into account. In particular, they now refer to some additional work on the co-evolution of behaviors and the environment, and they clarified the relevant timescale for their analysis. I would like to thank the authors for these changes. The paper can be published now.

---

## [Editor Report · Acceptance letter]

8 Sep 2023

PCOMPBIOL-D-22-01352R2 

Human-environment feedback and the consistency of proenvironmental behavior

Dear Dr Ecotière,

I am pleased to inform you that your manuscript has been formally accepted for publication in PLOS Computational Biology. Your manuscript is now with our production department and you will be notified of the publication date in due course.

With kind regards,

Anita Estes
